# Water flow in the active layer along an arctic slope – An investigation based on a field campaign and model simulations

Sebastian F. Zastruzny<sup>1</sup>, Bo Elberling<sup>2</sup>, Lars Nielsen<sup>1</sup> Karsten H. Jensen<sup>1</sup>

<sup>1</sup>Department of Geoscience and Natural Resource Management, University of Copenhagen, Copenhagen, 1350, Denmark

<sup>2</sup>Center for Permafrost (CENPERM), Department of Geosciences and Natural Resource Management, University of Copenhagen, Copenhagen, 1350, Denmark

Correspondence to: K. H. Jensen (khj@ign.ku.dk)

Abstract. As climate conditions change, the hydrological regime in the active layer is subject to change too. This influences the transport of solutes and the availability of nutrients, e.g. nitrogen particularly, along slopes. There is a lack of

- understanding the pathways and travel times of water and nutrients along slopes in discontinuous permafrost regions and how to scale changes along transects to the rest of the landscape. This study presents a comprehensive data set of a field site in Disko Island in Greenland aiming at constructing a hydrological model of the area. Data from automated weather stations, geophysical surveys, soil samples and soil sensors and tracer experiments are combined to describe the spatial variability in the field and to serve as input to a two-dimensional model (SUTRA) for simulating water and solute transport in the summer
- period. The model is calibrated and validated against volumetric water content and breakthrough curves of the applied tracers. Observed and simulated results suggest that the flow velocity in the active layer is directly influenced by annual precipitation patterns leading to water flow during the summer and rapid movement at the end of summer. Yearly travel times for the specific field site are simulated to be approximately 14 m/a and the highest peak velocities are most likely caused by preferential flow paths. The spatial heterogeneities linked to the frost topography seem to control the direction and
- velocity of flow. The observed discontinuous movement of a conservative tracer suggests that the movement of dissolved nitrogen compounds such as nitrate, being released along the slope in consequence of permafrost thawing, could possibly quickly influence nitrogen cycling at the end of the slope. This may trigger a feedback of climate changes in terms of increasing carbon sequestration due to additional plant growth in these otherwise nitrogen-limited Arctic ecosystems.

#### **1** Introduction

The active layer is the uppermost zone of the ground characterized by annual thaw-freeze cycles. The layer is in direct contact with the atmosphere and provides the basis of vegetation growth and is therefore an important part of permafrost-affected arctic ecosystems (Bonnaventure and Lamoureux, 2013). As a result of infiltration, a thin saturated layer is

- commonly formed at the base of the active layer and on top of the mostly impermeable permafrost table. Exchange with deeper groundwater is only possible if open taliks connect the active layer to deeper aquifers (Bense et al., 2012). The typical seasonal recharge pattern in arctic regions is characterised by two large events: Snowmelt at the beginning of summer and increased precipitation at the end of summer. In the main growing season in between, water storage and flow decline as evaporation exceeds precipitation except during short-duration precipitation events (Ireson et al., 2012).
- Permafrost inhibits mixing of supra-permafrost and sub-permafrost water (Tetzlaff et al., 2015), leading to longer travel times in the sub-permafrost aquifer and shorter solute travel times in the supra-permafrost aquifer (Frampton et al., 2011). The velocity and direction of flow in the active layer is controlled by the topography of the frost table, which acts as an impermeable barrier and therefore controls the flow pathways (Wright et al., 2009). The microtopography of the frost table creates a complicated flow networks that also lead to lateral flow (Atchley et al., 2015). Models capable of simulating the
- complex mechanisms occurring in cold climate environment have only recently emerged (Kurylyk et al., 2014) and are still under development.

An increase in average annual air temperature (IPCC, 2014) may lead to a thicker active layer by the end of the growing season as well as the formation of taliks and increased erosion which have implications for both the natural ecosystem and infrastructure (Jorgenson et al., 2010). Part of these impacts are directly linked to changes in the hydrological conditions

- (Bense et al., 2009) with profound changes in the permafrost hydrology, active soil moisture distribution and flow patterns (Bring et al., 2016). The deepening of the active layer will change the flow and transport pattern and typically lead to longer travel paths and times (Frey and McClelland, 2009;Frampton and Destouni, 2015). An ecological effect is related to the release and subsequent migration of inorganic nitrogen stored in the perennially frozen ground (Elberling et al., 2010) or from winter decomposition (Blok et al., 2016). These two processes occur at different times during the year and release
- nitrogen at different depths in the active layer. The production and transport of nitrogen in the active layer depends heavily on the depth where flow is occurring. With increased depth the effect of denitrification is reduced which increases the net export of nitrogen (Harms and Jones, 2012).

This phenomenon will particularly occur at sloping locations with high hydraulic gradient where the released nitrogen can be transported by water and supply nutrients to the low-lying wetlands in the arctic. Yano et al. (2010) applied <sup>15</sup>N to the

30 surface and their results confirmed that production and transport of nitrogen in the active layer depend on depth, with water playing the single most important role in the mobilization. They developed a conceptual model that emphasizes the differences between flow in the uppermost layer and flow in deeper layers where export of nitrogen is possible due to lack of

nitrogen sinks. Harms and Jones (2012) adopted this conceptual model, but applied <sup>15</sup>N at the base of the thaw layer by using a push-pull method. They also observed that nitrogen uptake occurred mostly during snowmelt in shallow soils with uptake rates significantly lower in summer and autumn, allowing for export of nitrogen from soils to lower lying wetlands or streams. Similar findings were achieved by using a water mass balance model and nitrogen flux model showing that during

- snowmelt a catchment can be a sink for water but at the same time be a source of nitrogen because of mobilization (Petrone et al., 2007). In a different study Harms and Ludwig (2016) showed that nitrogen concentrations are significantly related to water residence times, which differ in so called water tracks, channels found on arctic slopes draining an enhanced soil moisture zone (McNamara et al., 2008) and non-water tracks. They concluded that water tracks may augment the inorganic nitrogen reaching downslope ecosystems. Observations from long-term summer warming experiments in NW Greenland
- showed that warmer conditions accelerate biological processes of nitrogen transformation and wetter conditions lead to increased soil hydraulic conductivity, changed pathways of nitrogen transformation and extended periods of microbial activity (Schaeffer et al., 2013).

Several studies have emphasized the need for combined field experiments and modelling in order to address the knowledge gaps in permafrost hydrology related to water flow and transport in permafrost-affected sediments (e.g. Ireson et al.,

- 2012;Bring et al., 2016). The objective of the present study is to quantify water flow and transport mechanisms in the active layer along an arctic hill slope with particular focus on the potential transport mode and time for nutrients released upstream. The study is based on comprehensive hydrogeological and geophysical field experiments and observations in combination with numerical modelling of flow and transport in the active layer. Tracer experiments were carried out to obtain field evidence on the transport behaviour of an inert solute and hereby the migration of nutrients released in thawing permafrost
- on arctic slopes. The study is based on an underlying hypothesis that nutrients released from different parts of the landscape may be transported to the bottom of hill slopes and here trigger an ecosystem response due to the additional water and nutrients.

# 2 Description of Study Area

The study area is located on Disko Island in Greenland (69°27' N, 53°46' W) (Fig. 1a) on the border between high and low arctic conditions. The climate is classified as maritime low arctic with an annual average air temperature of -3.3°C for the period 1991-2004 and with maximum in July (mean monthly average 7.1°C) and minimum in February-March (mean monthly average -16°C) (Hansen et al., 2006b). For the observation period 1991-2011, the mean annual air temperature has increased by 0.2°C per year (Hollesen et al., 2015) now reaching near zero as the annual mean air temperature. Positive daily average air temperatures are typically between May and September (Hansen et al., 2006b). Precipitation as rain is 273 mm

with September being the wettest month; while precipitation as snow corresponds to 200 mm water equivalents (Hansen et al., 2006b).

The permafrost distribution on Disko Island is characterized as being continuous (Humlum, 1998) while recent temperature changes suggest more discontinuous permafrost (Hollesen et al., 2015). This is in line with the general trend of the climate

model predictions of the future climate of Greenland where permafrost temperatures are expected to increase which result in an associated increase in the thickness of the active layer (Daanen et al., 2011). The conditions of Disko Island are prone to change from a previous continuous permafrost area to a discontinuous area in consequence of changing climate.

The landscape is shaped by the Weichselian ice until 10.000 years ago and consists of glacial valleys eroded into basaltic layers and deposited tills. The retrieving glaciers left behind moraines of unsorted material, containing material of basaltic

- origin with a high percentage of iron (Pedersen et al., 2006). The field site is situated on a south-facing slope stretching 80 m in length between 110 m and 125 m a.s.l., on a moraine ("Pjeturssons Moræne") (Fig. 1b). The moraine stretches perpendicular in a valley ("Blaesedalen") and has a total extent of 1 km westwards from the site, where it is cut off by a river ("Rød Elve"). At the foot of the moraine is a wetland and 50 m further west of the profile under investigation, a lake. The wetland is separated by a small depression, which becomes a small creek after heavy rainfalls. The moraine continues north
- of the field site to a total altitude of 153 m a.s.l. A regional DEM of Disko Island is available from the WorldView-2 satellite.

The study area is selected as being as representative as possible for slopes in West Greenland and it exhibits typical features such as a diplotelmic activ layer, variable vegetation pattern and irregular occurrence of water flow. As located at the border between high and low arctic, the site is expected to be in the front line of climate change and the change from continous to

20 discontnous permafrost (Hansen et al., 2006a). The field site was established in August 2015 and the field investigations and data collection were carried out during the period August 4-27, 2015.

# 3 Materials and methods

#### 3.1 Climate variables

An automated weather station (AWS3) is located 300 m from the field site. Since 2012 data are supplied at 30 minutes 25 intervals for precipitation (Campbell Scientific, ARG1000), air temperature (Campbell Scientific, CS215), wind speed and – direction (Campbell Scientific, A100R) and soil water content at 5, 10, 20 and 30 cm depths (Campbell Scientific, SM300). An additional weather station (AWS2) located 3 km south-west from the site is measuring net radiation (Kipp & Zonen CNR4) and ground heat flux at 5 cm depth (Campbell Scientific HFP01SC-10) at 30 minutes intervals as well as snow depth (Campbell Scientific, SR50A).

To calculate evapotranspiration (ET) the Priestly-Taylor method (Priestley and Taylor, 1972) was used:

$$\mathbf{ET} = \frac{\mathbf{Q}_{\mathrm{E}}}{\mathbf{\rho}_{\mathrm{W}}\cdot\mathbf{L}_{\mathrm{v}}} = \frac{1}{\mathbf{\rho}_{\mathrm{W}}\cdot\mathbf{L}_{\mathrm{v}}} \, \boldsymbol{\alpha}_{\mathrm{PT}} \cdot \mathbf{f}(\boldsymbol{\theta}) \cdot \left(\frac{s}{s+\gamma}\right) \cdot \left(\mathbf{Q}_{\mathrm{N}} - \mathbf{Q}_{\mathrm{G}}\right) \tag{1}$$

where  $Q_E$  is the energy used for evapotranspiration,  $Q_G$  is the ground heat flux and  $Q_N$  is the net radiation both measured at 5 AWS2, s is the slope of the saturation vapour pressure curve dependent on temperature,  $\gamma$  is the psychometric constant,  $L_V$  is the latent heat of vaporization, and  $\rho_w$  is the density of water. The coefficient  $\alpha_{PT}$ , accounting for the ratio between potential evapotranspiration and equilibrium potential evaporation (Woo, 2012), is set to 1.26 as commonly assumed (Priestley and Taylor, 1972). The penalty function  $f(\theta)$  accounts for the reduction in ET for water limiting conditions. Here we assume a simple penalty function which varies linearly with the water content ( $\theta$ ), between the values for field capacity ( $\theta_{FC}$ ), and

10 permanent wilting point ( $\theta_{PWP}$ ), respectively:

$$\mathbf{f}(\boldsymbol{\theta}) = \frac{\theta - \theta_{PWP}}{\theta_{FC} - \theta_{PWP}}$$
(2)

The snow water equivalent (SWE) is calculated based on the formula by DeWalle and Rango (2008):

$$SWE = d_{snow} \frac{\rho_{snow}}{\rho_{water}}$$
(3)

where  $d_{snow}$  is the depth of snow,  $\rho_{water}$  is the density of water, and  $\rho_{snow}$  is the density of snow.

#### 3.2 Soil parameters

- Undisturbed and depth-specific volumetric soil samples (n=24) were collected from 11 locations along a transect along the hill slope and down to 50 cm depth (Fig. 2). The samples were kept in airtight containers for analysis in the laboratory. Soil properties were measured in the laboratory according to standard methods (Blume et al., 2011): carbon content (*C*) was measured by incarnation at 1350°C, porosity (*n*) and dry bulk density ( $\rho_{dry}$ ) were determined by gravimetric methods, and saturated hydraulic conductivity ( $K_S$ ) was measured using a flow cell. At the sample locations, the bulk heat capacity ( $c_w$ )
- and bulk thermal conductivity  $(K_T)$  of the soil were measured in situ with a KD2 (Decagon Devices) at 10 cm intervals.

#### 3.3 Soil moisture, temperature and electric conductivity

A total of 14 sensors (Decagon Devices 5TE) were installed at the site for measurements of volumetric water content ( $\theta$ ), soil temperature ( $T_{soil}$ ), and bulk electric conductivity ( $\sigma_b$ ). The sensors were installed at the top, middle and bottom of the transect at depths between 10 to 50 cm below ground (Fig. 2). The sensors were connected to a data logger (Campbell EM50) that monitored at 30 min intervals.

# **3.4 Geophysical Imaging**

Four multi-electrode electrical profiling (MEP) arrays were made to assess the presence and extent of frozen ground (Fig. 2). For data acquisition a Supersting R8 from AGI Geo was used. Electrodes were attached to pin electrodes that were placed in the ground to a depth of approximately 30 cm. Contact of the electrodes was ensured by a self-testing mechanism of the

- acquisition system. A data acquisition strategy based on a Wenner configuration was chosen because only limited horizontal variability of subsurface structures was expected and the primary focus was on delineation of horizontal layering of the subsurface (Ishikava, 2008). For the N-S profile 54 electrodes were installed along the line with a spacing of 1.5 m, yielding a profile length of 81 m, with 0 m profile distance located close to the foot of the slope. In order to allow for deeper penetration depth at the corners and to capture heterogeneity perpendicular to the N-S profile, three additional profiles W-E
- profiles were collected using 56 electrodes with a spacing of 2 m, yielding profile lengths of 110 m, relative to the N-S at the positions 7.5 m ('Bottom'), 31.5 m ('Middle') and 68.5 m ('Top'). The thickness of the active layer was measured using a stick probing at least every 2 m along the transect and used to validate/constrain the results based on the MEP independently.

The MEP data was processed and inverted using the software RES2DINV, which relies on an iterative smoothnessconstrained least squares inversion method (Loke, 2015). In order to obtain optimal results, noise, i.e. data points that showed quite strong and physically unrealistic deviation from neighbouring data points, were excluded manually from the modelling processes (Ishikava, 2008). For all profiles, convergence was reached after a maximum of five iterations. The convergence criteria used was that the relative change in root-mean-square value of the fit between observed and modelled data points changed by less than 5% or exceeded 5 iterations. The topography along the MEP profiles was extracted from a local digital elevation model (D'Imperio et al., 2017) and included in the modelling.

Supplementary to the ERT data, Ground Penetrating Radar (GPR) profiles were collected in order to provide independent geophysical constraints on the depth to the permafrost layer (Jørgensen and Andreasen, 2007;Gacitúa et al., 2012).

(4)

#### 3.5 Tracer Experiment

Salt (NaCl) was used as an inert solute tracer for mapping the flow pattern and transport times at the site. A total of 3.3 kg of salt was diluted in 10 l of water to obtain a NaCl solution that could be traced by measuring bulk electric conductivity  $\sigma_b$  of the soil. The tracer was added into a 30 cm wide trench dug to the permafrost base on August 6, 2015 at 16:00 at two positions, 35.8 m and 65.4 m from the foot of the slope. The tracer concentration was designed such that the plume migration could be monitored by the Decagon sensors, one at the top position and five at the middle position.

To capture the two-dimensional spreading of the tracer, the electric conductivity (EC) of the soil was additionally measured on a grid array with 50 cm spacing four times at dates August 13, 16, 21 and 27. The measurements were carried out using an auger (diameter 1 cm), which was inserted into the soil at the grid points until solid ground was hit. The lower 15 cm of

10 the core were diluted in 100 ml distilled water and the resulting EC of the suspension was measured with an EC-Meter (Hanna Instruments, HI99300) and back calculated to represent the EC of the solution phase. At the 50 cm grid the surfaceand permafrost topography was recorded by measuring the distance to a horizontal arbitrary plane above the surface and interpolating between data points.

#### 3.6 Numerical flow and transport modelling

- The model code SUTRA 2.2 (Voss and Provost, 2010) was used for analysing the data from the field site. The code is capable of simulating unsaturated fluid flow as well as heat and solute transport. Refreezing and thawing during the summer period is not considered. Furthermore, it is assumed that the active layer depth does not vary over the considered season and flow and transport are only simulated in a two-dimensional cross-section along the hill slope.
- Given these assumptions the governing equation for fluid flow reads:

$$\frac{\partial(nS_w)}{\partial t} = \nabla[K(\theta) \cdot \nabla \mathbf{h}] - \mathbf{q}$$

where n is the porosity,  $S_w$  is the water saturation, t is the time,  $K(\theta)$  is the unsaturated hydraulic conductivity, h is the hydraulic head and q is a sink term representing evapotranspiration.

For the retention and hydraulic conductivity functions we assume that the van Genuchten-Mualem relationships are applicable (Van Genuchten, 1980). For retention this relationship is defined as:

$$S_e = \frac{S_w - Sr}{1 - Sr} = \left(1 + (a_{VN} p_c)^{n_{VN}}\right)^{-1 + \frac{1}{n_{VN}}}$$
(5)

where  $S_e$  is the effective saturation,  $S_w$  the saturation,  $S_r$  is the residual saturation,  $\alpha_{VN}$  and  $n_{VN}$  are soil specific empirical values respectively, and  $p_c$  the capillary pore pressure. The relationship for unsaturated hydraulic conductivity  $K(\theta)$  is defined as:

5 
$$K(\theta) = K_S S_e^{0.5} \left[ 1 - \left( 1 - S_e^{\frac{n_{VN}}{n_{VN}-1}} \right)^{\frac{n_{VN}-1}{n_{VN}}} \right]^2$$
 (6)

where K<sub>s</sub> is the saturated hydraulic conductivity.

For solute transport simulations it is assumed that the tracer is inert, thus the solute transport is calculated based on the 10 advection-dispersion equation:

$$\frac{\partial(n\rho S_e c)}{\partial t} = \nabla[n\rho S_w \mathbf{D} \cdot \nabla \mathbf{c}] - \nabla(\nu n\rho S_w c)$$
(7)

where c is the concentration, D is the dispersion coefficient and v is the fluid velocity. The change in density based on the concentration of solute in the tracer liquid is calculated after McCutcheon et al. (1993).

# **4 Results**

#### 4.1 Soil parameters

All soil parameters were depth related reflecting an organic layer covering a lower mineral layer. Porosity and carbon content exhibited a clear difference between an upper organic and mineral soil layers (Fig. 3a). Average values for the porosity were 94.7 (± 2.6)% and 67.6 (± 11.4)% for the organic and the mineral layer, respectively, and for the carbon content 37.8 (± 3.7)% and 7.9 (± 5.1)%, where numbers in brackets represent the standard deviations. Figure 4 shows that the interpolated boundary varied between 12 and 28 cm below surface, with the active layer consisting entirely of organic material at some localities along the profile. Dry bulk density had values of 113.5 (± 58.5) g/l and 831.5 (± 308.7) g/l for the organic and the mineral layer, respectively. The high standard deviation for the mineral layer is a result of the high variability in clay, silt and sand fractions, which also is manifested in a high variability in hydraulic conductivity values. The

hydraulic conductivity values of the organic layer were fairly high 654.7 ( $\pm$  133.5) cm/d due to fibrous appearance of the

organic material and high porosity. The average hydraulic conductivity of the mineral layer was only slightly lower 279.2  $(\pm 279.7)$  cm/d, but on the other hand the variability for this layer was much higher (Fig. 3b).

The vegetation varied according to the distribution of soil characteristics, mainly related to differences in soil moisture, grain size distribution and active layer depth. For the upstream part of the slope, soils below the organic layer consisted primarily

of medium to coarse grained sand with an active layer depth >100 cm, while at the downhill locations soils were made up of silt or fine sand composition. Active layer probing at this location revealed thickness up to a depth of >120 cm.

#### 4.2 Moisture content variations

The volumetric water content  $\theta$  (Fig. 5) varied between 0.3 and 0.8 for all sensors except at location Mid<sub>3</sub>, where values of about 0.2 were measured until the August 28. Water contents were rather stable but increased abrupt due to rainfall events.

Stable values after September 29 are due to freezing conditions which is confirmed by soil temperature measurements. Very stable values are observed for sensor locations  $Bot_{W,D}$  and  $B_{W,S}$  from August 29 suggesting that saturation of the soil was obtained implying porosity values 0.4 and 0.6, respectively. The water saturated conditions at  $Bot_{W,D}$  are confirmed for the period from the August 8 until August 27 by direct observation in a dug trench, close to the location.

#### 4.3 Inversion results and conceptual model of the site

- A conceptual model of the field site was developed based on the topography and measurements of the active layer depth with a probe. The lower boundary of frozen material was estimated by electric resistivity tomography (ERT) that penetrated to a maximum depth of 20 m below the surface. The boundary between frozen and unfrozen sediment was found to correspond to a resistivity of approximately 1000 Ωm, based on comparison with the frost probing depth in the active layer (Fig. 6). Reflectivity observed in GPR data collected along the same line supports the interpretation of permafrost made based on
- frost probing depth (Fig. 4) and ERT data. Based on fitting of hyperbolas observed in the reflection image, an average velocity of 0.057 m/ns was estimated, which was subsequently used for migration and depth conversion (Neal, 2004). A reflection interpreted to represent the top of the permafrost layer is found at approximately 0.4 m depth at the downhill part of the transect (Fig. A2). This reflection is found a greater depth (about 0.5-0.6 m depth) from approximately 6 m profile distance until approximately 14 m profile distance from where is gradually becomes shallower until it reaches a depth of
- only about 0.3 m depth from approximately 18 m profile distance and onwards. It should be noted that from 7 to 14 m profile distance, relatively strong GPR reflectivity is observed 0.1-0.2 m above the interpreted permafrost depth based on the direct measurements. However, this apparent inconsistency is minor taking the vertical resolution of the GPR data, which is about 0.1 m, and inherent uncertainties regarding velocity estimation and depth conversion into account. Moreover, the slightly shallower reflectivity in this profile distance range may have a different origin. For profile distances larger than 68 m, the
- GPR data show relatively weak reflectivity indicating stronger attenuation, which is consistent with the drop/shift in

resistivity observed in this area in the ERT model and is interpreted as the existence of an open talik at the top of the field site (Fig. A3). Based on geomorphological features (lake and several palsas) at the foot of the field site, an open talik is likely at the end of the moraine. Observations at the field site and analysis of the soil samples indicate the existence of a typical two-layer system in the active layer with an upper high porous organic layer and a lower less porous mineral layer.

- The border between the two layers is estimated based on interpolation between the sample locations and using the carbon content as a proxy for identifying the boundary between the two horizons. Based on climatic conditions permafrost is likely to be found on higher locations on the moraine. This permafrost acts as an impervious cap that reduces the recharge to the sub-permafrost aquifer. By stopping vertical movement of water, a perched aquifer is formed at the bottom of the active layer in which water is subsequently routed downslope in the active layer. For the subpermafrost aquifer an expected low
- value for the hydraulic conductivity of the moraine till suggests little exchange of groundwater at depth, so that very limited flow can be assumed.

#### 4.4 Tracer Observations

The plumes of the applied tracer behaved differently at the top and middle locations (Fig. 7). At the top location, an elongated breakthrough curve was observed at sensor  $Top_{1.5}$  with a slowly rising breakthrough followed by an abrupt

- recession limb after August 29 in response to several days of precipitation. The average travel velocity until the peak concentration was 10 cm/d. At the middle location, an immediate response subsequent tracer application was observed at Mid<sub>1</sub>. Between August 13-18 the sensor failed but it seems that the concentration was fairly stagnant until August 29 when the concentration dropped rapidly in response to precipitation. Sensor Mid<sub>2</sub> reacted vaguely around August 12 and had a more abrupt reaction together with sensors Mid<sub>3</sub> and Mid<sub>4</sub> on August 28. Sensor Mid<sub>5</sub> did not show increase in concentration
- at any time. For the middle tracer the peaks appear approximately 2 days after heavy rainfall at the end of August. Based on the sensor responses at August 28 the average travel velocity was 170 cm/d (Fig. 7). The difference in migration behaviour at the top and middle locations is mainly due to differences in topography of the permafrost surface that was measured and interpolated based on a 50 cm grid at the two tracer sites. At the top location, the surface was homogeneous that allowed for flow and tracer transport predominantly in the direction of the slope. On the middle location, the permafrost
- topography was more heterogeneous with local depressions and channels implying a more non-uniform flow pattern (Fig. 8a). The difference in the surface configurations is influencing the shape of the plume as determined from the soil coring (Fig. 8b). For the top location, the tracer moved along the slope and overall formed an elongated and coherent plume while a more lateral-dominated transport is observed at the middle location. The heterogeneity of the permafrost surface created storage capacity in local depressions. In these depressions, the concentration decreased as they were slowly depleted
- by tracer but as new tracer from the application point entered the depressions again after rainfall the concentration increased as observed between August 21 and 27.

#### 4.5 Modelling Results

Based on the conceptual model described above, a two-dimensional numerical model is developed for the site (Fig. 6). The outline of the frozen part of the ground is not resolved as a sharp boundary by the employed ERT approach. The 1000  $\Omega$ m isocline of the ERT was used to define the outline of the frozen body, which is excluded from the model domain. At the

- edges of the parallel-running profile, the perpendicular ERT profiles were used to construct the boundary between frozen and unfrozen ground at depth. The combined knowledge from the stick probing and the GPR profile was used to construct the depth to the base of the active layer, and the interpolated results from the soil samples were used to establish the boundary between organic and mineral soil. Further, temporal variations in the active layer depth are not considered as observations have shown that changes in the active layer depth are modest and on average 3.5 cm between August 4 and August 21
- (Fig. 4). The observations suggest that maximum active layer depth is reached at the beginning of August confirming previous results by Humlum (1998). Back-freezing starts at exposed positions and depressions in the permafrost. For a field site with similar seasonal temperature variations, Boike et al. (1998) found variations in active layer depth between mid-July and mid-September to be less than 20%, which corresponds to 10 cm in the present case. Nevertheless, significant changes occurred close to the open talik where the depth changed 27 cm over 17 days (Fig. 4).
- An irregular mesh was defined for the model domain ranging from 0.1 m for the active layer to about 2 m for the subpermafrost aquifer (Fig. 6). The simulation period was constrained to the summer period, starting June 1 (Day 1) and ending October 1 (Day 122), during which period flow is active in the active layer. The time step was set to 2 hours, resulting in 1464 time steps. The first ten days served as a spin-up period. Initial conditions for June 1 were assumed to represent dry conditions and were thus set to permanent wilting point. However, subsequent infiltration of meltwater from the snowmelt
- brought the water content close to saturation implying that the assumptions regarding the initial conditions were not of significance. Fully saturated conditions were assumed in the sub-permafrost aquifer. The model was calibrated against observations of volumetric water content assuming that sensors with a lateral offset also represented the flow conditions in the transect. The calibration period was August 6 until September 24, where a decrease of soil temperature in the active layer was identified thus making frozen conditions possible. The root mean square error
- (RMSE) was used as an overall performance criterion:

$$RMSE = \sqrt{\frac{1}{N} \sum_{i=1}^{N} (OBS_i - SIM_i)^2} = ME^2 + STD^2$$
(8)

Calibration was performed in two steps. First a manual stepwise calibration was carried out to obtain values for interception 30 of rainfall by vegetation, fraction of melting snow infiltrating at the start of the simulation period, and the dynamic viscosity

of water. In a consecutive step automated calibration using PEST (Doherty, 2016) was carried out where the parameters porosity and saturated hydraulic conductivity were individually calibrated for three different soil types. Initial values (Table 1) were specified according to the laboratory measurements.

- The manual calibration showed that a higher interception factor and lower snowmelt infiltration generally improve the model performance. However, an interception factor above 0.2 deteriorates the dynamic behaviour of the water content in the active layer drastically, thus a value of 0.2 was used. The infiltration fraction of snow was set to 0.8, which is a rather high value. However, this parameter also partly accounts for water released upon thawing of the active layer and might not necessarily represent the actual infiltration of snowmelt water. The duration of snowmelt showed only limited sensitivity. The dynamic viscosity has large influence on the dynamics of the water content and best results were obtained for a viscosity at
- 10 temperature 10°C.

The automated calibration of the mineral layer resulted in high values for porosity and hydraulic conductivity that exceeded the values obtained from measurements on soil samples and higher than expected for a sand-silt sediment type. The resulting parameter values used in the model are listed in Table A1. The calibration resulted in a RMS of 0.175.

The calibration results are shown in Figure 9. For both the top and the bottom locations the simulated volumetric water content is mostly in between the observed water contents at lateral offset. Shortcomings in the model performance can be attributed to four causes:

1. The position of the sensor relative to the active layer may vary from location to location. Furthermore, the depth of the active layer may change. As the water table follows the frost table (Ireson et al., 2012) this can lead to divergent sensor readings.

20 2. For sub-arctic hillslopes the soil conditions are subject to high heterogeneity including the presence of preferential flow pathways (Carey and Woo, 2001). This leads to a high variability in soil variables and parameters even over very short distances, which is not accounted for in the model.

3. Field observations have documented that water flow on the top of the permafrost table is subject to local threedimensional effects, which cannot be accounted for in a two-dimensional cross-sectional model.

4. All temperature dependent effects are ignored; this does not only include variations in the active layer depth, but also effects such as cryo-suction and changing viscosity.

The calibrated model was validated against breakthrough curves of the inert tracer (Fig. 7). The measured values for bulk electrical conductivity  $\sigma_b$  were converted to electrical conductivity of the pore water  $\sigma_e$  based on the relation developed by Hilhorst (2000):

30

$$\sigma_e = \frac{\theta}{\theta_S} \frac{80.3 - 0.37 \cdot (T_{Soil} - 20)}{\varepsilon_b - \varepsilon_{\sigma_b = 0}} \sigma_b \tag{9}$$