# Peer review of "Water flow in the active layer along an arctic slope – An investigation based on a field campaign and model simulations"

_The Cryosphere, 2017_

## Referee Comment (RC1) · Anonymous Referee #1 · 28 Jun 2017

General comments:

This manuscript presents a study of hydrologic transport in the active layer of a slope in a permafrost environment on Disco Island, Greenland. It is based on a vast range of field data, including geophysical measurements, soil water content and electrical conductivity monitoring data, weather data, as well as a tracer experiment and numerical modeling. The objective is to quantify flow and transport mechanisms in the active layer.

The topic is very relevant as our understanding of transport in these systems is limited and our need to quantify these processes is growing as the Arctic warms and

permafrost thaws. The methods are also suitable for achieving this objective.

However, the manuscript (and study) lacks focus. Many methods are applied, but they are not all well motivated and described in the Methods section. This mainly concerns the geophysical methods, which are used to delineate permafrost at the study site. As the thickness of the active layer is monitored by manual probing at the site, it is not clear how the geophysical data add any information for the active layer. There is also no estimate of uncertainty at all for the geophysical data, which also make it impossible to judge if they provide any valuable information for deeper parts of the ground. Finally, as the focus of the study is transport in the active layer, I wonder if there is at all a need for the geophysical data in this study. If the Authors do a more thorough uncertainty analysis of the geophysical data, it could perhaps be used to add some information about hydrologic connectivity through taliks. Otherwise, I think the geophysical data could be removed from the study, yielding a more focused manuscript.

A related concern is the inclusion of the subpermafrost aquifer in the modeling. The configuration of the modeling domain is based on the electrical resistivity tomography data. However, it seems like all the transport results presented in the manuscript regard the active layer, and it is unclear if including the subpermafrost aquifer yielded any additional insights.

To summarize, I believe the manuscript (and study) could be much improved by focusing on its core methods, results and strengths. This might mean excluding some methods and data, but also highlighting more strongly what new insights were gained by this study. I hope that the Authors have the possibility to take the time needed to rework this manuscript and that my comments can be helpful in this endeavor.

Specific comments

Page 1, L19 What does "frost topography" mean? Do you mean the topography of the permafrost table?

Page 2, L2 First sentence: add that this regards permafrost areas.

Page 2, L11 Frampton et al. (2011) does not deal with transport times, however Frampton and Destouni 2015 does:

Frampton and Destouni (2015) Impact of degrading permafrost on subsurface solute transport pathways and travel times, Water Resources Research 51(9): 7680–7701.

Page 2, L14 Please specify what lateral means in this context.

Page 2, L14 Atchley et al. (2015) does not observe or study any "complicated flow netorks" or lateral flow per se, but acknowledges that this is a potentially important factor to consider when modeling heat and water in permafrost terrains. Perhaps there is a better reference for this (I don't know of any – sorry!) or the sentence could be slightly reformulated?

Page 2, L28 – page 3, L12 This paragraph, focusing on nitrogen transport, takes up a substantial part of the introduction, yet this study does not focus on nitrogen transport per se. I suggest focusing the introduction more on general transport processes in the active layer, as this is the focus of your study. The existing text could easily be rewritten and the same studies could be cited, but with a stronger focus on the transport aspects.

Page 2, L29 – L30 What "surface" did Yano et al. apply nitrogen to? This sentence needs some specifications.

Page 3, L20 Is this study designed to test this hypothesis? As you state that it is an hypothesis, readers will likely expect you to test this in the study, and I cannot see that you test for example ecosystem responses to additional nutrients and water. The idea that water and nutrients can be transported down a hillslope is not debated in our sciences, and your study design is more specific than that. Maybe you can reword a bit – "our underlying assumption"?

Page 4, L8 This sentence needs some rewording to make it clearer.

Page 4, L15 The existence of a DEM does not need to be mentioned here. If it was used in some way for the study, it can be mentioned in the Methods section.

Page 4, L28 Was snow depth also measured at 30 minute intervals? (Was it used for this study?)

Page 5, L7 What does the reference to Woo (2012) refer to? The alfa of 1.26 is taken from Preistley and Taylor (1972), right?

Page 5, L9 - L10 Where did you get the values for the wilting point and field capacity?

Page 6, L1-5 These sensors were in addition to the ones at the weather station? Were the sensors calibrated in anyway? What precision and accuracy do you expect?

Page 6, L6 – L25 The idea here, as I read it, is to use ERT and GPR to get some constraint of the distribution of frozen ground, that can be used for the setup of the numerical model. However, to do this, some information on the quality of these data is needed. What is the resolution of the ERT results for this Wenner array with 30 cm spacing, in the active layer as well as deeper in the ground? Does the ERT result give you any information about the AL, given this resolution and inherent uncertainty? What data is used for determining the resistivity boundary of frozen/unfrozen ground? What uncertainty do you have on this boundary value? Generally, the uncertainty of an ERT inversed model increases with depth. There are ways to identify areas of high uncertainty in the image (e.g. Oldenburg and Li, 1999; Marescot et al., 2003; Fortier et al., 2008), which I urge you to use if you want to say something about the thickness of permafrost.

Oldenburg, D. W. and Li, Y. G.: Estimating depth of investigation in dc resistivity and IP surveys, Geophysics, 64, 403–416, doi:10.1190/1.1444545, 1999.

Marescot, L., Loke, M. H., Chapellier, D., Delaloye, R., Lambiel, C., and Reynard, E.: Assessing reliability of 2D resistivity imaging in mountain permafrost studies using the depth of investigation index method, Near Surf. Geophys., 1, 57–67, 2003.

[Figure]

Fortier, R., LeBlanc, A. M., Allard, M., Buteau, S., and Calmels, F.: Internal structure and conditions of permafrost mounds at Umiujaq in Nunavik, Canada, inferred from field investigation and electrical resistivity tomography, Can. J. Earth Sci., 45, 367–387, doi:10.1139/e08-004, 2008.

Page 6, L7 Add that these are Electrical Resistivity Tomography (ERT) measurements.

Page 6, L26 – L27 A lot of information is missing on the GPR measurements. What antennas were used (frequency, shielding)? Were the measurements done in a common offset approach? What was the antenna distance? What was the sampling time window? Were traces stacked? How were the measurements done along the transect, every X cm or every X s, or other method?

How did you convert travel times to depth? What were the uncertainties, and the resolution? What uncertainties will stem from an assumption of homogenous velocities in the active layer, considering your observed variability in water content?

How was the GPR data processed? What software was used? Did you use any filters on the data, or time-zero correction?

Finally, considering the inherent uncertainties in GPR results and the fact that you have observations of active layer thickness from probing, what do these GPR measurements add to your studies? If you get the same information but with higher accuracy from probing, I would recommend excluding the GPR data from the manuscript.

Page 7, L8 What was the area of the grid?

Page 7, L14 - Page 8, L15 Please add a description of your model experimental setup, including information about the mesh, boundary conditions, initial conditions, hydraulic properties, simulated time, time stepping. Most of this information appears in the results section, but it should be here instead.

Figure 4 The organic layer thickness line varies also in places where there were no soil samples, but should be an interpolation between soil sample points (based on organic

carbon content). Can you explain this? I can find no mention that the active layer thickness was measured, in the methods section. I assume this was done manually with a probe (?). Please specify this in the methods section. The active layer also seems to be thicker on August 4th than on August 21st, which seems odd.

Page 9, L18 What is the uncertainty around this estimate? From just looking at figure 6, it seems that you could have a rather large range of resistivity values at the bottom of the active layer. You could perhaps provide a standard deviation as well as the 1000 Ohmm value. Do you expect the permafrost boundary to be at the same resistivity value also deeper in the ground, based on what you know about the ground substrate? What do you base this assumption on?

Page 9, L19 - When I look at figure A2 (the GPR image) I don't see any interpretation of the permafrost table that is based on the actual image, and I therefore cannot see how this image supports the frost probing depths. The frost probing seems to be the better data set here, and I cannot see how you need the GPR at all for what you do.

Page 10, L1 Does the active layer probing support this? Again, without any constraint on the uncertainty you cannot draw much conclusion here – but if you do some analysis of the uncertainty in ERT results you could actually say something about this potential talik.

Page 10, L2 How do observations of palsas and a lake suggest that there is a talik at the top (bottom?) of the studied moraine?

Page 10, L7 Where is higher, in relation to the talik (or taliks?) that is mentioned earlier in the paragraph?

Page10, L7-11 What do you actually know about the sub-permafrost aquifer? If this is all based on the ERT data, then you have a much stronger case if you do an uncertainty analysis on those results. However, this study is interesting enough if it focuses on only active layer transport. That could make a more focused paper and reduce some of the

uncertainty that stem from assumptions based on geophysical measurements.

Page 10 L21 Which location is this velocity for?

Page 11 This is text that should go in the methods section.

Figure 6 Is not a good figure for showing the model domain, as the actual domain is difficult to see behind the ERT model.

Page 11, L1-3 I wonder if you need the sub-permafrost aquifer in your model. What are the important information that you gain from including it? I believe you need to motivate better why it should be included in the model. It would be very cool if you could say something about exchange between supra- and sub-permafrost aquifers with your modeling.

Page 11, L15 What does it mean that the mesh ranges from 0.1 to 2 m? Is this a resolution of the mesh? Please specify this more clearly.

Equation 8 Explain this equation in the text.

Page 11, L22 Which soil moisture sensors were used for the calibration? Maybe I misunderstand something here, but you have sensors west and east of the transect which show rather different values for water content. Did you calibrate against a mean for those sensors?

Page 12, L11-13 Just a thought: Is RMS the best or only meaningful measure here? Would it be helpful to include something like the Nash-Sutcliffe model efficiency to evaluate the model fit? Just something to think about.

Page 12, L27 – Page 13, L7 This would fit better in the methods section.

Page 13, L9 "arrival correct times" should be "correct arrival times"?

Page 13, L15 – L23 As the results did not show very strong evidence (or focus) on the distribution of taliks, this does not feel like the most motivated issue to start the discus-

sion section with. Connection and exchange between sub- and supra-permafrost taliks are very interesting research topics, but, as I read this manuscript, this study focuses more on the active layer. I suggest to start this section with a discussion on that.

Page 15, L15 – Page 16, L13 The Conclusions section could be streamlined by moving some parts to the Discussions section. For example, the discussion on nitrate is more in depth here than in the Discussion section. I suggest you look through this section and make sure to move new lines of discussion to the Discussion section.

Finally, for inspiration - A couple of studies that I didn't see in your reference list but that might be of interest to You, came to my mind when I read Your manuscript:

Johansson, E., S. Berglund, T. Lindborg, J. Petrone, D. van As, L.-G. Gustafsson, J.-O. Näslund, and H. Laudon, 2015, Hydrological and meteorological investigations in a periglacial lake catchment near Kangerlussuaq, west Greenland – presentation of a new multi-parameter data set, Earth Syst. Sci. Data, 7, 93-108, doi:10.5194/essd-7-93-2015

Bosson, E., Sabel, U., Gustafsson, L. G., Sassner, M., and Destouni, G.: Influences of shifts in climate, landscape, and permafrost on terrestrial hydrology, J. Geophys. Res., 117, D05120, doi:10.1029/2011JD016429, 2012.

Technical corrections

Throughout the text the phrasing "active layer depth" occurs. As the active layer is a layer, it has a thickness rather than a depth. I suggest you change the wording from depth to thickness throughout the manuscript.

Page 2, L14 "a complicated flow networks"

Page 9, L23 "This reflection is found a greater depth. . ."

Page 14, L15 Perhaps both of the two occurring "generic" are not needed in this sentence?

---

## Author Comment (AC1) · 7 Jul 2017

General comments:

This manuscript presents a study of hydrologic transport in the active layer of a slope in a permafrost environment on Disco Island, Greenland. It is based on a vast range of field data, including geophysical measurements, soil water content and electrical conductivity monitoring data, weather data, as well as a tracer experiment and numerical modeling. The objective is to quantify flow and transport mechanisms in the active layer.

The topic is very relevant as our understanding of transport in these systems is limited and our need to quantify these processes is growing as the Arctic warms and permafrost thaws. The methods are also suitable for achieving this objective.

However, the manuscript (and study) lacks focus. Many methods are applied, but they are not all well motivated and described in the Methods section. This mainly concerns the geophysical methods, which are used to delineate permafrost at the study site. As the thickness of the active layer is monitored by manual probing at the site, it is not clear how the geophysical data add any information for the active layer. There is also no estimate of uncertainty at all for the geophysical data, which also make it impossible to judge if they provide any valuable information for deeper parts of the ground. Finally, as the focus of the study is transport in the active layer, I wonder if there is at all a need for the geophysical data in this study. If the Authors do a more thorough uncertainty analysis of the geophysical data, it could perhaps be used to add some information about hydrologic connectivity through taliks. Otherwise, I think the geophysical data could be removed from the study, yielding a more focused manuscript.

Reply:

We think that the geophysical data provides valuable extra information about the spatial variability of field site, which is an important issue in terms of describing the water flow in the area.

The GPR data is used to measure the active layer thickness in between the positions, where the stickprobing was done. As simple interpolation of the probing yielded results comparable in quality to the GPR data, the latter was used to represent the active layer thickness in the model domain. Thus, the GPR profiling is a valuable additional source of information to represent the natural spatial variability of active layer thickness.

The resistivity data provide valuable information on the lower boundary of frozen material as well as the presence of taliks.

All together, we consider the geophysical data to be important for the study. Consequently, we have kept the geophysical data as an integrated part of the paper. As suggested we will analyze the impact of uncertainty to explore if more information can be extracted from the data.

A related concern is the inclusion of the subpermafrost aquifer in the modeling. The configuration of the modeling domain is based on the electrical resistivity tomography data. However, it seems like all the transport results presented in the manuscript regard the active layer, and it is unclear if including the subpermafrost aquifer yielded any additional insights.

Reply: The existence of the open talik at the top of the slope creates a hydraulic connection between the active layer and the subpermafrost aquifer and may therefore impact the water balance and the flow velocities in the active layer. Although this dependency was qualitatively tested in a sensitivity analysis, currently we do not have field evidence, which can document the interaction between the supra- and subpermafrost aquifers. It is correct that the focus of the study is on flow in the active layer and also that most of the water flow takes place in this horizon. Nevertheless, we find it important to allow for the interaction between the two horizons in the model. Furthermore, the turn-around time for the numerical model is very fast and therefore inclusion of the subpermafrost aquifer is unproblematic.

To summarize, I believe the manuscript (and study) could be much improved by focusing on its core methods, results and strengths. This might mean excluding some methods and data, but also highlighting more strongly what new insights were gained by this study. I hope that the Authors have the possibility to take the time needed to rework this manuscript and that my comments can be helpful in this endeavor.

Reply:

We will consider focusing the manuscript as suggested. However, we believe that the strength of the study is that it is based on a variety of experimental and measurement

methods and we are therefore inclined to retain them in the manuscript. Nevertheless, the comments provided by the reviewer are extremely valuable for the revision of the manuscript.

Specific comments:

Page 1, L19 What does "frost topography" mean? Do you mean the topography of the permafrost table?

Reply:

We thank for the clarification. Yes we mean the topography of the permafrost table. However, we did refrain from using the term 'permafrost' table, as we used the thermal definition of permafrost throughout the manuscript, which therefore does not represent a physical state.

Page 2, L2 First sentence: add that this regards permafrost areas.

Reply:

Will be done.

Page 2, L11 Frampton et al. (2011) does not deal with transport times, however Frampton and Destouni 2015 does: Frampton and Destouni (2015) Impact of degrading permafrost on subsurface solute transport pathways and travel times, Water Resources Research 51(9): 7680–7701.

Reply:

Yes, we agree and apologize for this confusion.

Page 2, L14 Please specify what lateral means in this context.

Reply:

In this context lateral means flow perpendicular to the primary flow direction.

Page 2, L14 Atchley et al. (2015) does not observe or study any "complicated flow netorks" or lateral flow per se, but acknowledges that this is a potentially important factor to consider when modeling heat and water in permafrost terrains. Perhaps there is a better reference for this (I don't know of any – sorry!) or the sentence could be slightly reformulated?

Reply:

We will rewrite the paragraph to correctly refer to the results by Atchley et al. (2015).

Page 2, L28 – page 3, L12 This paragraph, focusing on nitrogen transport, takes up a substantial part of the introduction, yet this study does not focus on nitrogen transport per se. I suggest focusing the introduction more on general transport processes in the active layer, as this is the focus of your study. The existing text could easily be rewritten and the same studies could be cited, but with a stronger focus on the transport aspects.

Reply:

We will shorten the paragraph and focus the introduction more on solute transport.

Page 2, L29 – L30 What "surface" did Yano et al. apply nitrogen to? This sentence needs some specifications.

Reply: We will elaborate on the setup Yano et al. applied in their study.

Page 3, L20 Is this study designed to test this hypothesis? As you state that it is an hypothesis, readers will likely expect you to test this in the study, and I cannot see that you test for example ecosystem responses to additional nutrients and water. The idea that water and nutrients can be transported down a hillslope is not debated in our sciences, and your study design is more specific than that. Maybe you can reword a bit – "our underlying assumption"?

Reply:

We will reword this sentence.

Page 4, L8 This sentence needs some rewording to make it clearer.

Reply: We will reword this sentence.

Page 4, L15 The existence of a DEM does not need to be mentioned here. If it was used in some way for the study, it can be mentioned in the Methods section.

Reply:

We will mention the DEM in the method section and describe in which way it was used.

Page 4, L28 Was snow depth also measured at 30 minute intervals? (Was it used for this study?)

Reply:

The snow depth is measured at 30 minute intervals. However, we agree with the referee that the data is of no relevance for the present study and we will thus remove the sentence.

Page 5, L7 What does the reference to Woo (2012) refer to? The alfa of 1.26 is taken from Preistley and Taylor (1972), right?

Reply:

We agree that the reference at this point is not needed.

Page 5, L9 - L10 Where did you get the values for the wilting point and field capacity?

Reply:

Field capacity was based on repeated observation of soil moisture content after multiple days without rain at sensors close to the surface.

Wilting point was taken from the paper Moskal, T. D., Leskiw, L., Naeth, M. A., & Chanasyk, D. S. (2001), Effect of organic carbon (peat) on moisture retention of peat: mineral mixes. , Canadian Journal of Soil Science, 81((2)), 205-211, where wilting

point for peaty-sand sediment type is given.

This reference was by mistake not given in the manuscript.

Page 6, L1-5 These sensors were in addition to the ones at the weather station? Were the sensors calibrated in anyway? What precision and accuracy do you expect?

Reply:

The sensors were not specifically calibrated, but based on manufacturer specifications (Decagon Devices 5TE). The accuracies are according to the manufactorer's specification:

Apparent Dielectric Permittivity ($\varepsilon$a): $\pm$ 1 $\varepsilon$a (unitless) from 1 - 40 (soil range), $\pm$ 15% from 40 - 80 Soil Volumetric Water Content (VWC): Using Topp's equation: $\pm$ 0.03 m3/m3 ($\pm$ 3% VWC) typical in mineral soils that have solution electrical conductivity < 10 dS/m Electrical Conductivity (EC): $\pm$ 10% from 0 to 7 dS/m, user calibration required above 7 dS/m Temperature: $\pm$ 1°C

Page 6, L6 – L25 The idea here, as I read it, is to use ERT and GPR to get some constraint of the distribution of frozen ground, that can be used for the setup of the numerical model. However, to do this, some information on the quality of these data is needed. What is the resolution of the ERT results for this Wenner array with 30 cm spacing, in the active layer as well as deeper in the ground? Does the ERT result give you any information about the AL, given this resolution and inherent uncertainty? What data is used for determining the resistivity boundary of frozen/unfrozen ground? What uncertainty do you have on this boundary value? Generally, the uncertainty of an ERT inversed model increases with depth. There are ways to identify areas of high uncertainty in the image (e.g. Oldenburg and Li, 1999; Marescot et al., 2003; Fortier et al., 2008), which I urge you to use if you want to say something about the thickness of permafrost.

Oldenburg, D. W. and Li, Y. G.: Estimating depth of investigation in dc resistivity and IP

surveys, Geophysics, 64, 403–416, doi:10.1190/1.1444545, 1999.

Marescot, L., Loke, M. H., Chapellier, D., Delaloye, R., Lambiel, C., and Reynard, E.: Assessing reliability of 2D resistivity imaging in mountain permafrost studies using the depth of investigation index method, Near Surf. Geophys., 1, 57–67, 2003.

Fortier, R., LeBlanc, A. M., Allard, M., Buteau, S., and Calmels, F.: Internal structure and conditions of permafrost mounds at Umiujaq in Nunavik, Canada, inferred from field investigation and electrical resistivity tomography, Can. J. Earth Sci., 45, 367–387, doi:10.1139/e08-004, 2008.

Reply:

We agree with the referee that the quality and uncertainty of the geophysical data should be analyzed and described in more details. We will take advantage of the literature provided to strengthen this part of the manuscript.

Page 6, L7 Add that these are Electrical Resistivity Tomography (ERT) measurements.

Reply:

Will be done.

Page 6, L26 – L27 A lot of information is missing on the GPR measurements. What antennas were used (frequency, shielding)? Were the measurements done in a common offset approach? What was the antenna distance? What was the sampling time window? Were traces stacked? How were the measurements done along the transect, every X cm or every X s, or other method? How did you convert travel times to depth? What were the uncertainties, and the resolution? What uncertainties will stem from an assumption of homogenous velocities in the active layer, considering your observed variability in water content? How was the GPR data processed? What software was used? Did you use any filters on the data, or time-zero correction? Finally, considering the inherent uncertainties in GPR results and the fact that you have observations of active layer thickness from probing, what do these GPR measurements add to your

studies? If you get the same information but with higher accuracy from probing, I would recommend excluding the GPR data from the manuscript.

Reply:

The reviewer is right in pointing out that much information is missing on the GPR measurements. We will improve on this in the revised version.

Page 7, L8 What was the area of the grid?

Reply:

The top grid had the dimension 3m x 3m (9m2) and the middle one 3.5m x 2.5m (8.75m2). This will be mentioned in the revised version of the manuscript.

Page 7, L14 - Page 8, L15 Please add a description of your model experimental setup, including information about the mesh, boundary conditions, initial conditions, hydraulic properties, simulated time, time stepping. Most of this information appears in the results section, but it should be here instead. Figure 4 The organic layer thickness line varies also in places where there were no soil samples, but should be an interpolation between soil sample points (based on organic carbon content). Can you explain this? I can find no mention that the active layer thickness was measured, in the methods section. I assume this was done manually with a probe (?). Please specify this in the methods section. The active layer also seems to be thicker on August 4th than on August 21st, which seems odd.

Reply:

We will move the details on the numerical modeling to section 3 as suggested.

The interpolation of the organic layer thickness was done using kriging. In order to facilitate this, additional points were specified at the top and at the bottom of the domain representing maximum and minimum values for the carbon content, respectively. This led to variation of the organic thickness line in places with no measurements. We will

describe the procedure in more detail in the revised version.

The thickness of the active layer thickness was measured with a probe. This will be mentioned in the methods section.

As described on p. 11, L8-11 changes in the active layer thickness on the average is minor (3.5 cm). As also noted backfreezing has started between these two days in the depressions of the permafrost.

Page 9, L18 What is the uncertainty around this estimate? From just looking at figure 6, it seems that you could have a rather large range of resistivity values at the bottom of the active layer. You could perhaps provide a standard deviation as well as the 1000 Ohmm value. Do you expect the permafrost boundary to be at the same resistivity value also deeper in the ground, based on what you know about the ground substrate? What do you base this assumption on?

Reply:

We do expect that the permafrost boundary will be at the same resistivity value at depth since the underground settings are the same, which are unsorted moraine throughout the depth.

Page 9, L19 - When I look at figure A2 (the GPR image) I don't see any interpretation of the permafrost table that is based on the actual image, and I therefore cannot see how this image supports the frost probing depths. The frost probing seems to be the better data set here, and I cannot see how you need the GPR at all for what you do.

Reply:

This is a flaw in the figure. We will include the permafrost table in the revised version of the manuscript for reference.

Although the interpreted depth from the GPR is not directly used in the setup of the model it demonstrates the spatial variability of the settings and thus the complexity of

simulating water flow and transport in a permafrost affected hillslope. We consider the GPR data as supplementary data to the stick probing.

Page 10, L1 Does the active layer probing support this? Again, without any constraint on the uncertainty you cannot draw much conclusion here – but if you do some analysis of the uncertainty in ERT results you could actually say something about this potential talik.

Reply:

The active layer probing at that point showed non-frozen ground to a depth of 150 cm. Water samples from that location from a depth of 150 cm show a different isotopic signal than more surface near samples, which support the existence of a connection to a sub-permafrost aquifer.

Page 10, L2 How do observations of palsas and a lake suggest that there is a talik at the top (bottom?) of the studied moraine?

Reply:

The formation of palsa and pingos requires the presence of water (Woo, 2012). Especially pingos are often related to upwelling groundwater in permafrost areas, a phenomenon, which is termed 'pingo spring' (Gurney, 2000). Considering these geomorphological features, the size of the lake and that the permafrost in the area is classified as being discontinuous, we conclude that the lake provides thermal insulation, leading to the presence of a talik in the area.

Gurney, S. D. (2000). Relict Cryogenic Mounds in the UK as Evidence of Climate Change. In S. J. McLaren & D. R. Kniveton (Eds.), Linking Climate Change to Land Surface Change (pp. 209-229). Dordrecht: Springer Netherlands.

Woo, M.-K. (2012). Permafrost hydrology: Springer.

Page 10, L7 Where is higher, in relation to the talik (or taliks?) that is mentioned earlier

in the paragraph?

Reply:

Here we mean 'more uphill'. We will change this term in the next version of the manuscript.

Page10, L7-11 What do you actually know about the sub-permafrost aquifer? If this is all based on the ERT data, then you have a much stronger case if you do an uncertainty analysis on those results. However, this study is interesting enough if it focuses on only active layer transport. That could make a more focused paper and reduce some of the uncertainty that stem from assumptions based on geophysical measurements.

Reply:

We agree with the referee, that a uncertainty analysis could be beneficial and we will consider this, but as stated by the referee the focus of the study is on the active layer. We consider inclusion of sub-permafrost aquifer to be important for achieving good results for the active layer, however, the detailed shape and depth of the sub-permafrost aquifer is secondary in this regard.

Page 10 L21 Which location is this velocity for?

Reply:

The velocity of 170 cm/d refers to the middle location, directly at the lowest location of the active layer, observed for a short period of time.

Page 11 This is text that should go in the methods section. Figure 6 Is not a good figure for showing the model domain, as the actual domain is difficult to see behind the ERT model.

Reply:

We will move the numerical details to the methods section.

We will improve Figure 6 such that the model domain appears more clearly and we will consider splitting the figure in two separate figures.

Page 11, L1-3 I wonder if you need the sub-permafrost aquifer in your model. What are the important information that you gain from including it? I believe you need to motivate better why it should be included in the model. It would be very cool if you could say something about exchange between supra- and sub-permafrost aquifers with your modeling.

Reply:

Although the focus lies clearly on the dynamics in the active layer, the sub-permafrost is important to take into account when modeling the field site. Through the open talik at the top of the field site, water coming from the uphill active layer may enter the talik or remain in the active layer, depending on pressure in the sub-permafrost aquifer. Observations in the piezometers confirm the concept of this exchange. We will try to quantify this exchange between active layer and open talik based on the modeling in the revised version of the paper, to emphasize the necessity of the inclusion of the sub-permafrost aquifer.

Page 11, L15 What does it mean that the mesh ranges from 0.1 to 2 m? Is this a resolution of the mesh? Please specify this more clearly. Equation 8 Explain this equation in the text.

Reply:

An irregular mesh was defined for the model domain with the node distance ranging from 0.1 m for the active layer to about 2 m for the sub-permafrost aquifer.

Page 11, L22 Which soil moisture sensors were used for the calibration? Maybe I misunderstand something here, but you have sensors west and east of the transect which show rather different values for water content. Did you calibrate against a mean for those sensors?

Reply:

Where available the mean of the east and westwards located sensors was used as target for the calibration. In lack of other measures, the water content at the sensors (i.e. offset to the transect) were used as confidence intervals.

Page 12, L11-13 Just a thought: Is RMS the best or only meaningful measure here? Would it be helpful to include something like the Nash-Sutcliffe model efficiency to evaluate the model fit? Just something to think about.

Reply:

We will consider other measures such as the Nash-Sutcliffe coefficient for evaluating of the model.

Page 12, L27 – Page 13, L7 This would fit better in the methods section.

Reply:

Point taken.

Page 13, L9 "arrival correct times" should be "correct arrival times"?

Reply:

Will do. Page 13, L15 – L23 As the results did not show very strong evidence (or focus) on the distribution of taliks, this does not feel like the most motivated issue to start the discussion section with. Connection and exchange between sub- and supra-permafrost taliks are very interesting research topics, but, as I read this manuscript, this study focuses more on the active layer. I suggest to start this section with a discussion on that.

Reply:

Point taken.

Page 15, L15 – Page 16, L13 The Conclusions section could be streamlined by moving

some parts to the Discussions section. For example, the discussion on nitrate is more in depth here than in the Discussion section. I suggest you look through this section and make sure to move new lines of discussion to the Discussion section.

Reply:

We will make sure that discussion lines are in the discussion section and not in the conclusion section.

Finally, for inspiration - A couple of studies that I didn't see in your reference list but that might be of interest to You, came to my mind when I read Your manuscript: Johansson, E., S. Berglund, T. Lindborg, J. Petrone, D. van As, L.-G. Gustafsson, J.- O. Näslund, and H. Laudon, 2015, Hydrological and meteorological investigations in a periglacial lake catchment near Kangerlussuaq, west Greenland – presentation of a new multi-parameter data set, Earth Syst. Sci. Data, 7, 93-108, doi:10.5194/essd-7- 93-2015 Bosson, E., Sabel, U., Gustafsson, L. G., Sassner, M., and Destouni, G.: Influences of shifts in climate, landscape, and permafrost on terrestrial hydrology, J. Geophys. Res., 117, D05120, doi:10.1029/2011JD016429, 2012.

Reply:

We will take advantage of the above studies when revising the manuscript.

Technical corrections Throughout the text the phrasing "active layer depth" occurs. As the active layer is a layer, it has a thickness rather than a depth. I suggest you change the wording from depth to thickness throughout the manuscript.

Page 2, L14 "a complicated flow networks"

Page 9, L23 "This reflection is found a greater depth. . ."

Page 14, L15 Perhaps both of the two occurring "generic" are not needed in this sentence?

Reply:

We will adapt the technical corrections in the revised version.

We very much appreciate the insightful comments from the reviewer, which will be extremely useful for the revision of the manuscript.

---

## Referee Comment (RC2) · Anonymous Referee #2 · 15 Jul 2017

This paper presents 1) extensive data on a particular permafrost site, 2) performs a tracer experiment, and 3) builds a 2D model to simulate transport of a conservative tracer by calibrating to volumetric soil water content. The central focus of this work attempts to describe the control of advective flow in the active layer just above the permafrost. This is largely motivated by the need to understand nutrient transport and vegetation response. In all, there seems to be a very impressive amount of work here and I applaud the effort to describe all that work! Unfortunately it is not presented in an organized fashion, and connection between the measured field data, simulated tracer experiment, and nitrogen transport is not well described. The unorganized presentation style makes it very hard to understand the work, the assumptions used, and what

the central findings are, as well as how the details presented support the conclusions of the work. While I believe these subjects to be of high value to the cryosphere community, this manuscript is not suitable for publication in its current form. I recommend major revisions, and would encourage the authors to focus on what the measured field data and accompanying modeling are actually quantifying, rather then attempt to make conclusions regarding the fate of nitrogen and plant response. I see opportunity for scientific contribution if the authors focus on how these environments affect the transport of the conservative tracer, which in my opinion is a wide-open topic of cryosphere research. However, the authors must better justify their assumption of a stationary ALT. This is a critical weakness of this work; even if the change of ALT during the simulation is small, it could have a large impact because of the unsaturated conditions result in a smaller water table above the permafrost and lower lateral hydraulic transmissivity relative to a fully saturated ALT. At the very least a discussion on how a temporally variable ALT can affect transport is warranted.

Major Comments: 1) It is a very un-organized manuscript. Model assumptions are made, and the justification for those assumptions are not addressed until much later in the manuscript. Methods appear in the results section and the introduction and abstract are misleading.

2) For example it seems from the introduction, abstract and conclusion that this work intends to address how nitrogen transport may trigger feedbacks on increasing carbon sequestration due to additional plant growth, and to describe using a tracer experiment and model how nitrogen transport happens. Unfortunately, the methods and results do not get at nitrogen transport. Rather it only attempts to simulate a conservative tracer, which I believe could actually be a valuable science contribution unto itself. But the understanding of how the conservative tracer would move through the heterogeneous and dynamically adjustable ALT are not well described, and it is unclear how these processes contribute to the observed and simulated results.

3) Specifically, nitrogen and most nutrients are not a conservative tracer that is applied

to the landscape on a single event as represented in both the field and modeling experiment. Rather it is affected by biogeochemical processes and is continually applied to the system as a result of the nitrogen cycle. No discussion was provided regarding the difference of the experiments and how nutrient transport will behave, which are substantial. However, I would recommend abandoning the discussion and perhaps some of the motivation of nutrient transport altogether because the jump between the conservative tracer experiment and simulation to nutrient transport is too big and lot of necessary process understanding of the conservative tracer transport in these environments is lost in this manuscript. Here, I believe it is vital to fully understand the complexity of transport in the cryosphere before moving on to how biogeochemical reactants behave.

4) The transport of the conservative tracer in the heterogeneous environment is not well described nor is there much recognition of the copious work in hydrology regarding transport in heterogeneous environments, which could suggest a lack of critical understanding regarding all the processes that may affect the observed results. These processes include advective mixing and dilution, multiple breakthrough times from preferential flow ect. (e.g. Rajaram & Gelhar, 1995; Cirpka & Kitandidis 2000; Kung et al., 2000). Or how might highly transient adjective flow affect the tracer spreading, dilution and transport (Goode & Konikow, 1990), or diffusive flow into and out of low permeable areas that cause late time tailing affects (Schumer et al., 2003; Zhang et al., 2007), especially during times of low flow and unsaturated conditions. Here I suggest that the authors address these processes in the interpretation of both the site experiment and simulated results. Clearly, advective flow from snowmelt and precipitation pulses will drive system response, but I imagine preferential flow in the heterogeneous soils to be very important here. As is the transient boundary conditions from precipitation pulses, snow melt and an even moderate deepening in the ALT, that will result in different advective directions at different times. Goode, D. J., & Konikow, L. F. (1990). Apparent dispersion in transient groundwater flow. Water Resources Research, 26(10), 2339-2351. Cirpka, Olaf A., and Peter K. Kitanidis. "Characterization of mixing and dilution

in heterogeneous aquifers by means of local temporal moments." Water Resources Research 36, no. 5 (2000): 1221-1236. Kung, K-JS, T. S. Steenhuis, E. J. Kladivko, T. J. Gish, G. Bubenzer, and C. S. Helling. "Impact of preferential flow on the transport of adsorbing and non-adsorbing tracers." Soil Science Society of America Journal 64, no. 4 (2000): 1290-1296. Rajaram, Harihar, and Lynn W. Gelhar. "Plume‐Scale Dependent Dispersion in Aquifers with a Wide Range of Scales of Heterogeneity." Water Resources Research 31, no. 10 (1995): 2469-2482. Schumer, Rina, David A. Benson, Mark M. Meerschaert, and Boris Baeumer. "Fractal mobile/immobile solute transport." Water Resources Research 39, no. 10 (2003). Zhang, Yong, David A. Benson, and Boris Baeumer. "Predicting the tails of breakthrough curves in regional‐scale alluvial systems." Groundwater 45, no. 4 (2007): 473-484.

5) While the authors attempt to justify not resolving a moving ALT, how it affects the results is not concisely described, which in my opinion would be a extremely interesting contribution to the current state of the science. This is an especially odd boundary condition to assume because the authors partly motivate this work by citing the importance of variable ALT (see Lines 17-23 on Page 2). Here a small deepening of ALT of 4cm could still have a considerable affect to the results especially considering that ALT is approximately 40cm deep and is not fully saturated, which amounts to a greater then 10% change in subsurface storage of the saturated zone. This assumption probably needs to be tested or better yet the question 'how much does even a moderate change in ALT affect transport times?' needs to be answered.

Minor Comments

Page 1, Lines 8-11, This is a bit excessive motivation for an abstract, and may mislead the reader into thinking that the paper is dealing specifically with nitrogen transport in the ALT. Really the paper is about modeling the water flow in the ALT, perhaps it is better to narrow the motivation to what the paper actually discusses.

Page 1, Line 18. What are the units m/a? Meters annually? If so, it probably better to

use meters / year.

Page 1, Lines 20-23: This seems more of a discussion point and not a result of the paper. How specifically does this paper address nitrogen speciation, transport and subsequent vegetation growth. It is not clear that it does, and there for these lines should be removed from the abstract. Also, on line 22, 'climate changes' should be 'climate change.'

Page 2 Lines 7-8: Is it typical for all permafrost sites to have late summer precipitation or just the site investigated here? I would expect variability, and that some permafrost sites do not get late summer precipitation. You may need to specify that this only pertains to your site, and therefore this statement would be better served in the site description section rather then the introduction.

Page 2 Lines 13-14: While I agree that microtopography creates complicated flow networks that affects how lateral flow occur, unfortunately Atchley et al., 2015' is not an appropriate citation for this statement as the modeling in that paper was all 1D vertically, and the inclusion of any lateral flow discussion in that paper was only offered as a possible reason calibrations were not always successful. Painter et al., 2016, demonstrates 2 and 3D flow in polygons would be a good alternative citation as would Helbig et al., 2012, which is a more observationally based conceptualization of lateral flow and microtopography.

Painter, Scott L., Ethan T. Coon, Adam L. Atchley, Markus Berndt, Rao Garimella, J. David Moulton, Daniil Svyatskiy, and Cathy J. Wilson. "Integrated surface/subsurface permafrost thermal hydrology: Model formulation and proof‐of‐concept simulations." Water Resources Research 52, no. 8 (2016): 6062-6077.

Helbig, Manuel, Julia Boike, Moritz Langer, Peter Schreiber, Benjamin RK Runkle, and Lars Kutzbach. "Spatial and seasonal variability of polygonal tundra water balance: Lena River Delta, northern Siberia (Russia)." Hydrogeology Journal 21, no. 1 (2013): 133-147.

Page 2, Lines 14-16: It is true that models capable of simulating freeze thaw in the unsaturated zone is a relatively new development, but there are now several examples of models capable of simulating this. For example ATS - Painter et al., 2016, GEOtop - Endrizzi et al., 2014, JULES – Chadburn et al., 2015, Pflowtran – Karra et al., 2014. In fact I think SUTRA (used here) is also capable of simulating freeze thaw (Shemin et al., 2011). So the question I have is how does this work further that development, especially since freeze thaw dynamics are not simulated in this work?

Endrizzi, S., Gruber, S., Dall'Amico, M., and Rigon, R.: GEOtop 2.0: simulating the combined energy and water balance at and below the land surface accounting for soil freezing, snow cover and terrain effects, Geosci. Model Dev., 7, 2831–2857, doi:10.5194/gmd-7-2831-2014, 2014. Chadburn, S., E. Burke, R. Essery, J. Boike, M. Langer, M. Heikenfeld, P. Cox, and P. Friedlingstein. "An improved representation of physical permafrost dynamics in the JULES land-surface model." Geoscientific Model Development 8, no. 5 (2015): 1493-1508.

Painter, S. L. and Karra, S.: Constitutive model for unfrozen water content in subfreezing unsaturated soils, Vadose Zone J., 13, 4, doi:10.2136/vzj2013.04.0071, 2014.
 Shemin, McKenzie J, Clifford Voss, and Qingbai Wu. "Exchange of groundwater and surface‐water mediated by permafrost response to seasonal and long term air temperature variation." Geophysical Research Letters 38, no. 14 (2011).

Page 2, Lines 21-22: "The deepening of the active layer will change the flow and transport pattern and typically lead to longer travel paths and times (Frey and McClelland, 2009;Frampton and Destouni, 2015)." This statement and particularly the work of Framption and Destouni 2015, is why I think transport in permafrost and the dynamic ALT is so interesting. The freeze thaw cycles and the deepening of the ALT create additional complexity into transport systems that are not fully understood. Given that freeze thaw cycles are not simulated nor is the deepening and closing the ALT in this work, how does the work presented here contribute to understanding these complexities? From the methods, results, and discussion sections I cannot discern the contribution of

this work with regards to a dynamic impermeable zone, like permafrost. Page 3 Lines 18-20. It seems that that a goal of this study is to use a conservative tracer as a proxy for nutrient transport, specifically nitrogen. However, most nutrients are affected by biogeochemical process. There should be a discussion regarding the difference between a conservative tracer and a constituent that is affected by biogeochemical processes.

Page 4, Line 11: Replace 'retrieving' with 'retreating'

Page 4, Line 18: Replace 'activ' with 'active'

Page 7, Line 7: "To capture the two-dimensional spreading of the tracer", I suspect you mean 2D in the horizontal spreading rather then 2D in the vertical? Perhaps rephrase.

Page 7, Lines 16-18: If the transport is fast enough, then the assumption of a constant ALT and no thawing and refreezing holds. But what is the length of the simulation? In the above paragraph it states that a tracer was applied and observations were taken over the length of 21days. Was the modeling the same length and did the ALT change much over that time? Figure 7, shows both observed and simulated data beyond September 14th , so more then 40days? The length of observational data and especially the simulation need to be explicitly stated.

Page 8, Line 1: specify that $\alpha$VN and nVN Are van Genucthan parameters.

Page 8 Line 18, All parameters where depth related? Do you mean where a function of depth?

Section 4.1: Soil parameters. Why not summarize this data in a table?

Figure 3: It would be my preference to summarize this data in a table rather then figure. But just a suggestion. Also, why is the plot Dry Density / Hydraulic conductivity?

Page 11, Lines 8-17: Here is the explanation of why thawing and freezing was not simulated, seems out of place. Consider reorganizing your observations and model configuration to be closer to Page 7, Lines 16-18.

[Figure]

Page 11, Lines 15-21: This seems like it should be in the methods section.

Figure 4: The two sets of gray lines are hard to read, could a different color be used? This is somewhat important as the difference, or lack there of between the two lines are perhaps used is justification of not modeling freeze thaw processes.

Page 11-13: Results largely cover calibrating to volumetric water content. Yes calibrating to volumetric water content was mentioned in the abstract but judging from the introduction this is not exactly what I thought the paper would be about. I thought it would be more about simulating transport. Only Page 12 lines 27 to Page 13 line 12 actually take about validating the simulated tracer break through to observed tracer. From the abstract and intro, I thought the paper would be more about tracer transport. And yes I see why calibrating to volumetric soil moisture content would help get a model that produced correct-ish transport times, but shouldn't the model be constrained by hydraulic head ore pressure? As it is that pressure gradients and hydraulic conductivity is most going to affect advective transport.

Section 4.5. This section is mostly devoted to model calibration. A lot can be learned from model calibration to observed results and so this could be very useful information and help shape the conceptual model. So I was happy to see this discussion. However in addition to the above comment about better introducing the calibration process, I think this section also needs to be re-organized. I would suggest linking the volumetric soil water content calibration to the tracer experiment in some way. Also, shortcomings were identified on page 12 lines 15-29. How might understanding these short comings change your conceptual or numerical model?

Page 12 Line 20: Specify what flux your are referring to in, "...indicate that the flux in..."

Page 14 Line 8-10: Awkward phrasing. How about "Rapid flow is only possible only if recharge exceeds a certain threshold allowing for built-up of a saturated zone at the base of the active layer that interconnects the local depressions.

Section 5.2 Solute transport in the Active Layer: This section should be the focus of the work, but seems to lack rigor, perhaps due to the unfocused writing. A am not sure what the take home message is hear. Is it that precipitation causes increases in advectve flow? That seems pretty obvious, but what about all the other complexities of transport in heterogeneous media? At this point there is very little mention of preferential flow, and virtually no mention of how the combined affects of transient adjective flow, diffusive flow into low permeable areas; especially during times of low flow, and preferential flow paths can affect results like this. I would encourage the authors to concisely describe the processes at work here and explicitly relate them to the observed and simulated results.

Page 14 Line 20: "…this effect…" Precisely what effect is being referred to here? Several ideas/process are being described here and it is very hard to follow what effect is being discussed. The writing is in need of clarification and conciseness. Given that this is a complicated subject it is critical that the writing is clear.

Page 14 Line 22-23: "The velocity by which the active layer thaws controls thereby where the rapid movement occurs:" Very awkwardly phrased, I am not at all sure what is said here.

Page 15, Line 4-5: Again, Atchley et al., 2015 is not a suitable citation here.

Page 16, Lines 1-4: This discussion is out of place in the conclusions, but may fit in the discussion section. However, I would rethink the whole nitrate motivation and discussion altogether. While I agree that transport of nutrients is important, and understanding advective flow through the ALT will go a long way in describing nutrient transport, this manuscript shows very little work that can be applied directly to the transport of nitrogen. 1) nitrogen is not a conservative tracer and is affected by biogeochemical processes. 2) Nitrogen release is not in a pulse fashion that happens in the same manor a tracer test occurs. Rather its release into the subsurface will happen at varying levels over the entire growing/decomposing season.

Page 16, Lines 7-9: "This study confirms that a complex topography with ridges and depressions leads to lateral flow and retention in small depressions until an increase of the water table allows for overspill of the tracer." I think this statement is more aligned with this work then the nitrogen transport motivation. However, the paper as written doesn't clearly show this result, because the writing is often distracting from what I thought or hoped the central work was, advective flow in the ALT.